# Steps towards a Dislocation Ontology for Crystalline Materials

Ahmad Zainul Ihsan[1], Danilo Dessì[2,3], Mehwish Alam[2,3], Harald Sack[2,3], and Stefan Sandfeld[1,4]

[1] Forschungszentrum Jülich, Institute of Advanced Simulations – Materials Data Science and Informatics (IAS-9), Germany
{a.ihsan, s.sandfeld}@fz-juelich.de

[2] FIZ Karlsruhe – Leibniz Institute for Information Infrastructure, Germany
{danilo.dessi, mehwish.alam, harald.sack}@fiz-karlsruhe.de

[3] Karlsruhe Institute of Technology, Institute AIFB, Germany

[4] RWTH Aachen University, Faculty 5, Germany

**Abstract.** The field of Materials Science is concerned with, e.g., properties and performance of materials. An important class of materials are crystalline materials that usually contain "dislocations" – a line-like defect type. Dislocation decisively determine many important materials properties. Over the past decades, significant effort was put into understanding dislocation behavior across different length scales both with experimental characterization techniques as well as with simulations. However, for describing such dislocation structures there is still a lack of a common standard to represent and to connect dislocation domain knowledge across different but related communities. An ontology offers a common foundation to enable knowledge representation and data interoperability, which are important components to establish a "digital twin". This paper outlines the first steps towards the design of an ontology in the dislocation domain and shows a connection with the already existing ontologies in the materials science and engineering domain.

**Keywords:** Dislocation Ontology · Ontology Design

## 1  Introduction

Dislocation – one-dimensional lattice defects in crystalline materials – are responsible for plastic deformation of, e.g., metals or semiconductors, and play an important role concerning mechanical properties such as the hardening behavior. Since the dislocation had been postulated in the 1930s by Orowan [1], Taylor [2], and Polanyi [3], significant work has been dedicated to visualize dislocations through different microscopy techniques and to predict the evolution of dislocations by various simulation types. While dislocations are directly related to aspects on the atomic scale, in many situations the idealized representation of the dislocation as mathematical line is sufficient or even preferred. To understand the behavior of materials, however, aspects from both scales need to be

considered. This makes the knowledge representation difficult, even though this has not been perceived as a major research hindrance in the past.

During the past years, data-driven approaches made new methods and tools for analyzing and understanding the evolution of dislocation systems possible [4–6]. Similarly, the whole field of Materials Science and Engineering (MSE), a parent domain of the specialized domain of dislocations, is currently also undergoing almost disruptive change. This change brings simulations and experiments together, enabled by data-driven/data science approaches, and ultimately makes the digital transformation in the field of MSE possible [7–9]. One of the key enabler of the digital transformation is the Digital Twin (DT) [10]. DT is a digital representation of a real physical asset that represents the relevant features of the real asset within a model. For instance, to represent the whole process the experiment, a material digital twin could be created. The relevant features assigned from the experimental data are then represented by one or several simulation models. As one of the desired effects, each model within the DT may re-use the data resulting from another model, e.g., in a multi-scale simulation approach. To make the DT possible, new ways of handling research data and data annotation are needed, in particular to be able to use the data in an interoperable way and such that the descriptions of both the real asset and its virtual representation are unambiguous. Knowledge representation through a formal symbolic representation, i.e., through an ontology, is able to support data handling and to enable interoperability between related domains. Such an approach allows the domain knowledge to be represented by a set of axioms that can be understood by the machine. Furthermore, such a representation is explicit, i.e. the meaning of all concepts are defined, and it is shared by a common consensus.

In this work, first steps towards formalizing the knowledge of dislocations together with the relevant details concerning the crystallography are introduced. Emphasis is put on representing the dislocation geometry, utilizing semantic formalization using an ontology. We start by designing and modeling a formal definition of crystalline materials in terms of the underlying crystallography including slip plane and slip direction. The former is the plane to which the motion of the dislocation is generally constrained, the latter is the direction along which plastic deformation takes place. Those should be explicitly described along with the idealization of dislocation as a mathematical line and other required details of crystalline materials.

The paper is organized as follows. Section 2 presents related work of materials science domain ontologies and points out the existing gaps. Section 3 describes the physical aspects of dislocations along with the proposed ontology. Finally, Section 4 concludes and sketches the envisioned future work.

## 2  Related ontologies in the field of Materials Science and Engineering

Over the past three decades, a number of groups were involved with explicit conceptualization of the knowledge of their own MSE domains by means of on-

tologies. One of the earliest materials ontology is the Plinius ontology [11]: the authors developed an ontology for ceramic materials that covers the conceptualization of chemical compositions ranging from the single atom to complex chemical substances. In the "Materials Ontology" of Ashino et al. [12], the authors developed a detailed structure of materials information consisting of substances, process, environment, and properties of the materials. The recent development of materials digitization [13] has shown an initial step to describe the process-structure-property relation of a material defined by an ontology that represents the workflow applied to the specimen, e.g. heat treatment, specimen extraction, and tensile testing.

Another ongoing effort to establish semantic standards that apply at the highest possible level of abstraction, under which all conceivable domain ontologies can be subsumed and interoperated, is the European Materials and Modelling Ontology (EMMO)[1] developed by the European Materials Modelling Council (EMMC)[2]. It provides a common semantic framework for describing materials, models, and data with the possibility of extension and adaptation to other domains. For instance, the authors in [14] demonstrated the application of EMMO in the domain of mechanical testing, and in [15] the authors addressed the challenges that arises when level-domain ontologies are combined with EMMO by ontology alignment. Unfortunately, EMMO currently does not contain many sub-domains, which also includes the domain of dislocation.

Apart from work related to EMMO, efforts also have been made to represent the domain knowledge of crystal structures. The authors of MatONTO [16] have developed a Crystalline Structure Ontology as a sub-module of MatONTO by means of mapping and re-engineering the terms from the Crystallographic Information File (CIF) dictionary [17]. Since CIF and the alternative CIF2 standard are published and distributed under the umbrella of the International Union of Crystallography[3], it serves as a *de facto* standard for this community. The authors of CIF in [18] also have further developed the STAR/CIF Ontology that is written in the mathematical symbolic script language called dREL [19].

Recent development work concerning the Materials Design Ontology (MDO) [20] resulted in an ontology that covers the field of materials design, e. g., with regards to ab-initio calculations. Within MDO, a crystalline structure ontology module is developed to represent information of the atomic structure of materials. Yet, all these ontologies do not explicitly represent the physical and conceptual aspects of the crystal structure that are directly related to the representation of defects in crystals, i.e., lattice or slip planes, lattice or slip directions; furthermore, geometrical details of dislocations can – so far – not be represented. In summary it can be concluded that even though significant progress has been made concerning the ontology design in a number of related domains, it becomes clear that a level-domain ontology of dislocations in crystalline materials is still missing. Furthermore, there is still a gap between the existing crystal structure

---

[1] https://emmo-repo.github.io
[2] https://emmc.eu
[3] https://www.iucr.org/

ontologies in terms of the representation of crystalline defects. Therefore, in this work, the first steps towards ontology design of dislocations in crystalline materials are taken. For this, the terms from MDO will be connected and reused, especially these terms that describe the crystal structure, e.g., the atom entity and the lattice to enable data interoperability between domains.

## 3    Description of the Physical Domain and Ontology Design

### 3.1    Representation of Crystalline Materials and Line Defects

Before the ontology is designed, the most relevant concepts and notions for crystalline materials and line defect are now described. This is by far not a complete introduction; for more information the reader is referred to, e.g., [21, 22].

Crystalline materials are characterized by a periodic arrangement of the constituent atoms (see Fig. 1, left for an example). The representation of crystal

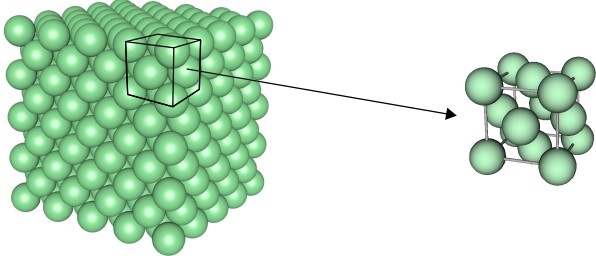

**Fig. 1.** An aggregate atoms in crystalline materials and crystal structure of face-centered cubic with one unit cell.

structures consists of the *lattice* together with a *motif*: the lattice is a mathematical concept of an infinite, repeating arrangement of points in a space (3D), in a plane (2D), or on a line (1D), in which all points have the same surrounding and coincide with atom positions. The motif (or base) consists of an arrangement of chemical species which in the real crystal can be atoms, ions, or molecules. One can now identify a smallest pattern of atoms which upon repetition along all spatial directions would again cover the whole structure: this pattern is the *unit cell*, shown as the black parallelepiped in Fig. 1. The edge lengths of this cell define the three *lattice parameters*. As shown in Fig. 2, to fully characterize the unit cell altogether six parameters are needed: three lengths, $(a, b, c)$ and three angles $(\alpha, \beta, \gamma)$. Based on these parameters, unit cells are often classified into a *crystal system*. There are seven distinct crystal systems, often ordered according to the increasing symmetry: cubic, tetragonal, orthorhombic, hexagonal, trigonal, triclinic, and monoclinic. E.g., the cubic structure has $a = b = c$, $\alpha = \beta = \gamma = 90°$ and the monoclinic structure has $a \neq b \neq c$, $\alpha = \gamma = 90°, \beta \neq 90°$.

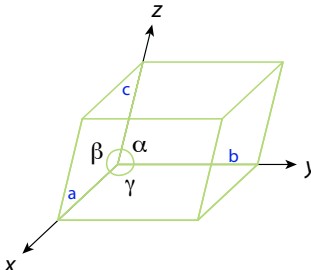

**Fig. 2.** The definition of the geometry of a unit cell requires three lengths and three angles.

Based on these definitions one can now define *lattice points*, *lattice directions*, and *lattice planes* as shown in Fig. 3: the lattice consists of a set of lattice points (which are the points where atoms or molecules are located), a lattice direction or lattice vector is a vector connecting two lattice points, and a lattice plane, forms an infinitely stretched plane (characterized through a plane normal) that cuts through lattice points such that again a regular arrangement of lattice points in the plane occurs.

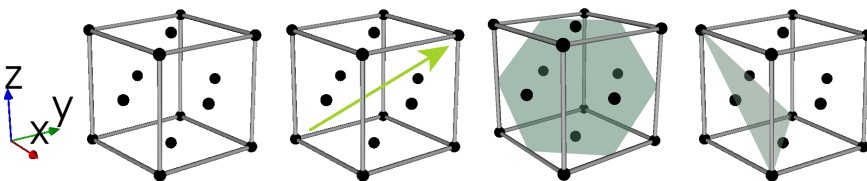

**Fig. 3.** From left to right: a "face-centered cubic" unit cell where the black points are lattice points and denote atom positions; a possible lattice direction where the vector connects to lattice points; and two different lattice planes.

### 3.2   Description of Linear Defects

Real crystalline materials generally don't have perfect order of atoms – at least not everywhere. Typically, a piece of material contains a large number of crystalline defects where the local order of the crystal structure is destroyed. Such a defect might be a point defect (e.g., a missing atom) or a line defect/dislocation (a strongly localized, tube-like region of disorder, which contains in the center the highly disordered dislocation core). While in general the notion of "defect" has a somewhat negative connotation, it is exactly this deviation from the perfect structure that result in important, e.g., electrical, mechanical, or thermal properties. In the following the one-dimensional defect type is considered; other defects will be discussed in a follow up publication.

In the context of plastic deformation, a dislocation is defined as the boundary of a slipped area within which atoms are displaced by the size of an elementary unit translation given by the so-called *Burgers vector*. Yet, the question arises on which granularity level a dislocation should be defined? One can define it in terms of displaced atoms, or – as will be done in the following – one can take a *mesoscopic* view where individual atoms are no longer visible and the tube-like defect "region" is in fact reduced to a mathematical line. The motion of this line through the crystalline material is constraint to a specific crystallographic plane. Details of the motion are determined by details from the atomic scale which is why the full crystallographic information still is required, even though the "defect region" is now idealized as mathematical line. These two different levels of detail requires particular attention when it comes to designing the dislocation ontology.

The motion of the dislocation is constrained to a specific crystallographic plane, called the *slip plane*. Within the slip plane, there are specific *slip directions* along which plastic deformation occurs, given by the so-called Burgers vector. A *slip system* is then defined as the set of slip planes with the same unit normal vector and the same slip direction is defined. Thus, the slip system is fully determined by the unit normal vector and the slip direction or the Burgers vector (where the latter is not a unit vector).

On the mesoscale, the mathematical dislocation object as shown in Fig. 4 is a directed curve that has a start point and an end point. The local line orientation changes along the line while the Burgers vector is constant for each point of the line. Lastly, since the dislocation is a directed curve it has a line sense which unlike the local line orientation is a property of the whole line.

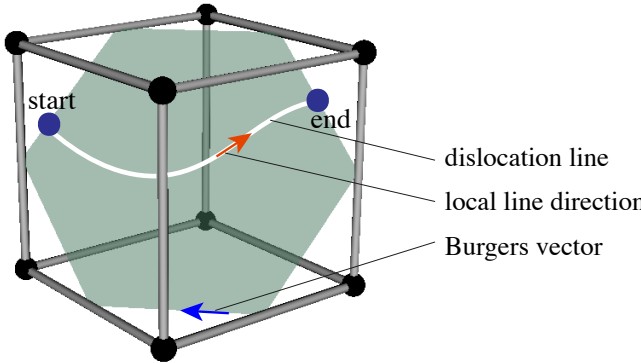

**Fig. 4.** Depiction of dislocation on the mesoscale as a mathematical object that has start and end points. The object is characterized by the Burgers vector and the line sense.

### 3.3   Ontology Design

The ontology design of dislocations in crystalline materials begins with the conceptualization of the crystal structure, followed by that of dislocations in crystalline materials. For this, the Crystal Structure module and the Dislocation module are developed. As shown in Fig. 5, the classes from `MDO`[4]`:Lattice` and `MDO:Occupancy` defining the lattice concept and the motif concept as an arrangement of chemical species in the crystal structure, respectively, are connected and reused.

The MDO contains only the class `MDO:Lattice` but does not contain any terms such as lattice point, lattice direction, and lattice plane. These will be added in our ontology. The `MDO:Lattice` class is refined such that each lattice individual consists furthermore of a `UnitCell` which is defined through the six lattice parameters (`LatticeParameterLength` and `LatticeParameterAngle`).

Given that the dislocation moves on a preferred plane, the slip plane, the slip plane concept needs to be described first. As shown in Fig. 5, `CrystalStructure` has the `SlipPlane` (a subclass of `LatticePlane`) and `SlipSystem`. Furthermore, on the `SlipPlane` there are specific directions called `SlipDirection` along which plastic slip happens and which is a subclass of the `LatticeDirection`. Finally, each `CrystalStructure` individual has a `SlipSystem` consisting of normal direction (`SlipPlaneNormal`) and `SlipDirection` of the respective slip plane. The dislocation module as shown in Fig. 6, represents the `Dislocation`

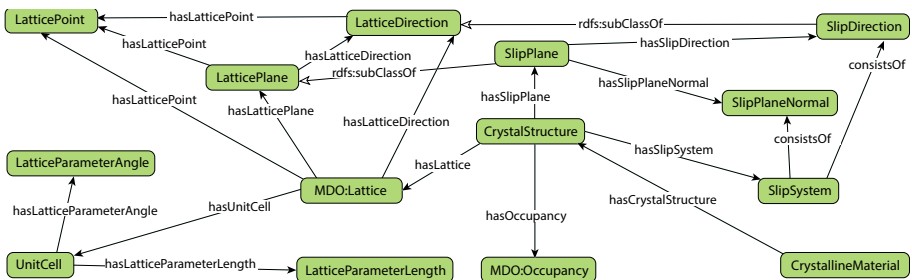

**Fig. 5.** Concepts and relations in the crystal structure module.

as subclass of a `CrystallineDefect` from which all crystalline defects (zero-dimensional up to three-dimensional defects) are derived. The `Dislocation` relates to `BurgersVector` and `LineSense`. Furthermore, since the dislocation only moves on a preferred type of plane, a relation between the `Dislocation` and its `SlipPlane` is defined. The `CrystallineMaterial` together with the `SlipPlane` are the link to the crystal structure module.

---

[4] Materials Design Ontology (MDO). https://w3id.org/mdo

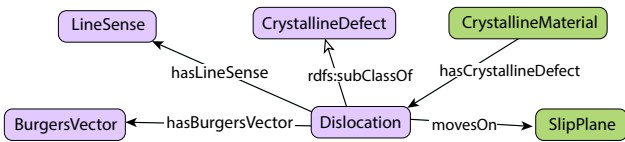

**Fig. 6.** Concepts and relations in the dislocation module.

## 4   Conclusion and Outlook

In this paper, steps towards an ontology design of dislocations in crystalline materials are presented. This has been done by developing a modular ontology of crystal structure and dislocation where common classes from the MDO have been reused and adapted. Future work will focus on the refinement of the above introduced concepts, so that further aspects and concepts from the materials science domain are included. Examples of this extension are the differential geometrical representation of dislocations as curved lines, consideration of the Bravais lattice but also other types of defects (e.g., point defects, grain boundaries). In addition, to extend the interoperability between domain ontologies especially within the MSE community, ontology alignment into EMMO also would be a very worthwhile undertaking.

## 5   Acknowledgements

AI and SS acknowledge financial support from the European Research Council through the ERC Grant Agreement No. 759419 MuDiLingo ("A Multiscale Dislocation Language for Data-Driven Materials Science") and Helmholtz Metadata Collaboration (HMC) within the Hub Information at the Forschungszentrum Jülich.

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
