# OpenReview forum: "Steps towards a Dislocation Ontology for Crystalline Materials"
_eswc-conferences.org/ESWC/2021/Workshop/SeDiT — SeDiT 2021 Oral_

### Official Review · AnonReviewer2 · 2021-03-19
**Interesting but fails to describe the connection with digital twins**

**Rating:** 6
**Confidence:** 4

**Review:**

This article describes the initial steps towards the development of an ontology in the dislocation domain. It seems to be a yet unaddressed area of knowledge that requires from an ontology to formally represent it. Although the need for an ontology is well motivated and the proposed steps are sound, it fails to show the role of the ontology with regards to the digital twins (which is a core part of the workshop). I think it is essential that authors clearly explain this issue. Other than that, I would like to see the ontology once it is developed and published. I propose this article as a position paper.

Some minor comments:
- The decision to reuse an existing ontology (in this case, MDO) was good and aligned with the ontology engineering best practices. I wonder why authors opted for MDO instead of MatOnto. Maybe you could elaborate on that a little bit.
- The placement of Figures 1 and Figure 3 is odd (in the middle of a paragraph)

---

### Official Review · AnonReviewer1 · 2021-03-28
**worth exploring but not much relation with digital twins**

**Rating:** 6
**Confidence:** 4

**Review:**

The paper presents the first steps towards an ontology for dislocation in crystalline materials. The design requirements (that is the physical domain and dislocation information) is very well explained. A comprehensible state of the art is provided and the paper is in general easy to read and follow.


Main motivations for the decision given about this paper are:

-- Even though digital twins are mentioned, it is in a shallow way being the relation between the ontology and its applicability to the digital twins domain insufficient.

-- No example of use of the ontology is provided

-- It is not clear the methodology used to build the ontology. Are authors following ontological engineering methodologies? Why the decision for using MDO rather than another ontology?

-- There is no link to the ontology so that it can be reviewed. Also, how is it planned to be shared/documented and manintained?

---

### Official Review · AnonReviewer3 · 2021-04-05
**Interesting - more detail requested on the domain ontology**

**Rating:** 6
**Confidence:** 5

**Review:**

This article proposes 'steps towards' a dislocation ontology for crystalline materials. It does so by proposing an ontology that is meant to be linked to the EMMO ontology. It includes particularly a number of classes that allow to describe the relevant geometry for the representations of lattices (slipPlane etc.). While I understand the aim and challenges, I have a number of concerns:

- I find the paper limited, in particular in terms of application value. When the paper finally starts outlining the research results and proposal (Section 3.3), it is stopped, and the conclusion follows. I had expected more details and tests for validation. I think that is also needed. So I suggest reducing content in pages 2 and 3, which is not so relevant at this point of the work, because not used anyway, and expanding Section 3.3, perhaps even in a separate section (see also last comment below).
- The article proposes an ontology, but there is no example represented with this ontology, which leaves validation of this ontology poor and results very uncertain. Please, include an example dataset (simple) that shows how to use the ontology on a particular crystalline material.
- If possible, please make the ontology available for testing.
- I understand that many people argue that 'placing a domain ontology under an upper-level ontology enables interoperability', but reality is very different. In reality, specialised content as the one in this paper is just interoperable within its own domain, which is also fine. So, I suggest to focus much less on this 'interoperability' aim, or 'linking to other ontologies', as outlined in Section 2; and instead consider more closely the data that you are meant to handle (Section 3). When developing the ontology starting from the domain of discourse, as you do in Section 3, a much more valuable result can emerge.
- The article assumes OWL as a language. Yet, there is no real argument or consideration of 'why'. In fact, RDF graphs (and labelled graphs in general) are known to be poor at representing ordered lists, incl. 3D geometric data and numerical data streams. This particular domain likely includes a lot of numerical analysis, which integrates very poorly with RDF technology. So... how will you deal with this? Even if the core concepts could be represented in an OWL ontology, I think you need to integrate with other technologies that are better in data-intensive operations (e.g. numerical analysis). This needs to be considered early enough, I would assume, to avoid the risk of ending up with a result that you cannot use.
- Section 3.1 and 3.2 can be shortened, because much of the domain knowledge in those sections will not match the knowledge and expertise of the audience in this workshop. Figure 2 and 3 can (arguably) be kept out, allowing to add more content in Section 3.3. For example, in Section 3, I can use an explanation of the colours used in Fig. 5 and 6. Also, I would really like more detail here. For example, are you including domain and range restrictions for your properties (e.g. what is the domain and range for 'hasLatticePoint'); does the presence of subClassOf restrictions in Fig. 5 with regular object properties not lead to circular effect (is that intentional?); which data properties do you include in the ontology; how do you represent 3D space, with WKT strings?; what about coordinate systems, do you need to represent those as well (how?); and so on. Many workshop participants will value these details.
- What is the added value for the domain experts of the alignment with MDO? I don't think they gain a lot with that?

Good luck with the article and research :)

---

### Decision · Program_Chairs · 2021-04-08

Accept (Oral)